# Response of Industrial Warm Drainage to Tide Revealed by Airborne and Sea Surface Observations

Donghui Zhang [1], Zhenchang Zhu [2,*], Lifu Zhang [1,3], Xuejian Sun [1], Zhijie Zhang [4], Wanchang Zhang [1], Xusheng Li [5] and Qin Zhu [6]

1 Aerospace Information Research Institute, Chinese Academy of Sciences, Beijing 100094, China
2 Guangdong Provincial Key Laboratory of Water Quality Improvement and Ecological Restoration for Watersheds, Institute of Environmental and Ecological Engineering, Guangdong University of Technology, Guangzhou 510006, China
3 Key Laboratory of Oasis Eco-Agriculture, Xinjiang Production and Construction Corps, Shihezi University, Shihezi 832003, China
4 School of Geography, Development & Environment, University of Arizona, Tucson, AZ 85721, USA
5 National Key Laboratory of Remote Sensing Information and Imagery Analyzing Technology, Beijing Research Institute of Uranium Geology, Beijing 100029, China
6 Southern Marine Science and Engineering Guangdong Laboratory (Guangzhou), Guangzhou 511458, China
* Correspondence: zhenchang.zhu@gdut.edu.cn; Tel.: +86-1580-0924-643

**Abstract:** Maintaining the balance between power station operation and environmental carrying capacity in the process of cooling water discharge into coastal waters is an essential issue to be considered. Earth observations with airborne and sea surface sensors can efficiently estimate distribution characteristics of extensive sea surface temperature compared with traditional numerical and physical simulations. Data acquisition timing windows for those sensors are designed according to tidal data. The airborne thermal infrared data (Thermal Airborne Spectrographic Imager, TASI) is preprocessed by algorithms of atmospheric correction, geometric correction, strip brightness gradient removal, and noise reduction, and then the seawater temperature is inversed in association with sea surface synchronous temperature measurement data (Sea-Bird Electronics, SBE). Verification analyses suggested a satisfied accuracy of less than about 0.2 °C error between the predicted and the measured values in general. Multiple factors influence seawater temperature, i.e., meteorology, ocean current, runoff, water depth, seawater convection, and eddy current; tidal activity is not the only one. Environmental background temperature in different seasons is the governing factor affecting the diffusion effect of seawater temperature drainage according to analyses of the covariances and correlation coefficients of eight tidal states. The present study presents an efficient and quick seawater temperature monitoring technique owing to industrial warm drainage to sea by means of a complete set of seawater temperature inversion algorithms with multi-source thermal infrared hyperspectral data.

**Keywords:** airborne remote sensing; seawater temperature monitoring; tidal activity; thermal infrared remote sensing; industrial warm water drainage





## 1. Introduction

Variations of seawater temperature near nuclear power plants under different tidal conditions are essential to verify the mathematical and physical simulation results of thermal drainage before the design and construction of the nuclear power plant. The extent and scope of the impact of a large amount of heat brought by the warm drainage (cooling water) from the nuclear power plant on the seawater temperature near the seashore area [1]; moreover, such an impact can change the basic species of the marine environment and have an unexpected impact on biodiversity and the stability of the ecosystem [2,3], which needs to be investigated for the safe operation of a nuclear power station. Remote sensing

technology has been a research hotspot to evaluate the impact of industrial activity on ecological balance [4]. Three core issues, i.e., the use of multi-source remote sensing thermal infrared data, the coupling relationship between warm drainage diffusion and marine environment [5,6], and the analysis of seawater temperature diffusion factors based on specific cases were explored by scholars under the guidance of the safe operation of a nuclear power station and coastal environmental protection.

Firstly, thermal infrared remote sensing sensors onboard satellites are mounted on airborne and Unmanned Aerial Vehicles (UAV), and some handheld sensors layout over the sea surface can constitute an effective "Space-Air-Ground" monitoring system for surface temperature observations. The wavelength of the thermal infrared sensor is between 8–12 μm, just located in the atmospheric window. Thermal Infrared Sensor (TIRS) [7], Moderate-resolution Imaging Spectroradiometer (MODIS) [8], Visible Infrared Imaging Radiometer (VIIRS) [9], The Sea and Land Surface Temperature Radiometer (SLSTR) [10], and Spinning Enhanced Visible and Infrared Imager (SEVIRI) [11] are common satellites in-orbit for Global Land Surface Temperature (LST) observations. Airborne thermal infrared sensors carried on the aircraft can obtain thousands of square kilometers of high-definition data with a resolution of 1 m in a few hours, making up for the lack of satellite transit time [12], which has been widely used in mineral exploration, environmental assessment and disaster monitoring represented by Thermal Airborne Spectrographic Imager (TASI) of Canadian ITRES [13]. Recently, due to the low cost of UAV thermal infrared technology, its application in the field of environmental monitoring has drawn more and more attention [14]. The representative technology is the Zen XT Series thermal imaging sensor of China DJI. Portable thermal infrared instruments are widely used in various fields. Multi-source data from these "Space-Air-Ground" monitoring systems can provide a reliable estimation of water temperature alone or produce multi-scale data to improve the accuracy of monitoring. Several categories of algorithms for the estimation of water temperature can be sub-divided, i.e., a temperature and emissivity separation algorithm, a multiple linear regression algorithm, and a neural network algorithm [15], according to the availability of the labeled data [16]. Recently, portable Unmanned Aerial Vehicles (pUAVs) are emerging as a promising alternative for the high spatial-resolution monitoring of a coastal hydro-environment, with the key advantage that they can be implemented with a frequent schedule to perform on-demand monitoring [17]. Common UAV thermal infrared sensors include Specim fx series [18], DJI Zenuse H20T [19] and Guide sensmart TMS7108A [20], which can provide 8–12 μm thermal infrared data. Regulators are increasingly using UAVs to fill these gaps in both spatial and temporal data resolution, causing the coarse resolution of Satellite and manned aircraft-based remote sensing [21].

Secondly, it is recognized that the warming of water, whether caused by humans or natural factors, leads to changes in the original bio-species of the ocean. Such change is controllable in a short period of time but may cause dramatic changes in the surrounding biota in long run [22]. Certain correlations exist between water temperature and tides [23], gravity [24], biology [25], meteorology [26], storms [27], industrial production, seasons [28], seabed topography [29], basement properties [23], . . . etc. In recent decades, relevant studies achieved remarkable progress under the context of global warming [30].

Thirdly, the key physical characteristics of coastal ecosystems, such as the health status of aquatic vegetation, surface water temperature, types of aquatic organisms and so on, have been identified effectively with the advancement of remote sensing technology [31]. The monitoring of oil film distribution, ocean bottom topography [32], coastline changes [33], suspended solids distribution [34], and seashore erosion [35] are all related to reliable estimation of seawater temperature. The monitoring of sea surface temperature is usually difficult spatially as opposed to temporally. Water temperature diffusion dynamic study is difficult due to limited information, and only the sea surface temperature can be obtained with remote sensing technology [36]. Additionally, the comprehensive understanding of the influence of wave, tide, season, and other factors on sea surface temperature is as of yet insufficient.

Given the present progress status of the aforementioned study issues, the present study aims to further examine the integration of thermal infrared remote sensing sensors mounted on airborne vehicles, as well as some handheld sensors layout over the sea surface to acquire a complete set of multi-scale remotely sensed information for sea surface temperature field retrieval. The study site of industrial warm water drainage chosen for the research purpose is in the tropical region near the seashore of Hainan province, as shown in Figure 1. Thermal infrared data derived from satellite and airborne flight have been preprocessed and the information extracted [4,14]. The final goal of this study is focused on exploring the main factors that govern the heat dissipation effect of industrial warm drainage in association with tidal, seasonal, and spatial data to answer the following questions: (1) to what extent and degree has the warm drainage caused a warming effect on the sea? (2) What are the differences in warm drainage diffusion dynamics under different tides? (3) How different are the warm drainage cooling effects in different seasons? (4) What is the controlling factor resulting in seawater temperature rising with regard to warm drainage?

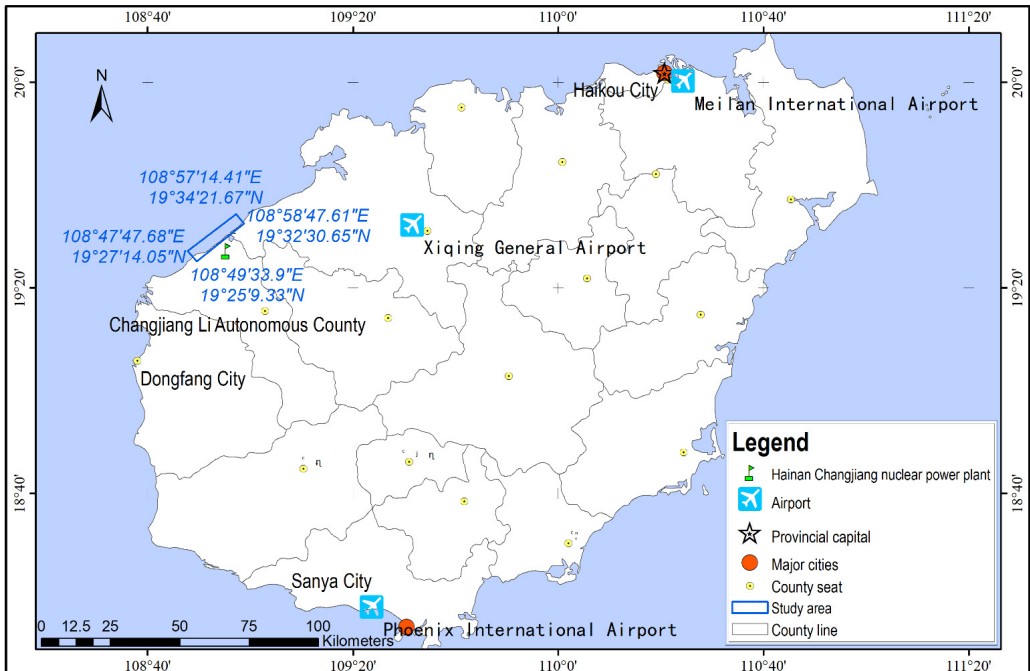

**Figure 1.** Geographic location of the study area in Hainan province with administrative division and airports available for present study.

## 2. Materials and Methods

### 2.1. Study Area

#### 2.1.1. Background

The Hainan Changjiang nuclear power plant is located on the seashore close to Beibu Gulf about 2 km southwest of Xingang village, Haitou Town (geographical coordinates range from 108°53′41″E~108°54′54″E to 19°28′0″N~19°29′10″N), Changjiang Li Autonomous County, Hainan Province. A study area was designed slightly larger than the seawater temperature rising area (rectangle marked area with blue as shown in Figure 1) simulated numerically with the nuclear power plant's drainage outlet as the center, referring to the nuclear power plant's temperature drainage area [4]. The Hainan Changjiang nuclear power plant is designed and constructed to the planned power generation capacity of $4 \times 650$ MWe. The once-through cooling system is adopted for cooling the discharged warm water of the power plant by taking the seawater as the circulating cooling water source. Warm drainage from the power plant after cooling by seawater is again discharged to the sea and diffused with the tidal current to the nearby seashore area (G et al., 2008). In this process, the warm drainage temperature is raised up to 8 °C at the outlet of the

power plant. The water intake and drainage flow of the once-through cooling system is about 81.59 m$^3$/s. The total flow of water intake and drainage of the planned capacity is approximately 163.18 m$^3$/S.

### 2.1.2. Coastal Environment

According to the data of Guangdong Water Resources and Hydropower Research Institute since 1950, the annual mean temperature in the sea area nearby the power plant is about 24.9 °C, and the coldest month in a year is averaged approximately 18.8 °C in January according to the meteorological record. The temperature fluctuated slightly over whole seasons of the year with a maximum difference of about 10.5 °C. The hottest season appears in June and July of the year with an annual mean maximum temperature of about 28.8 °C. The annual mean minimum temperature is about 21.8 °C and usually appears in January, while the lowest temperature recorded in January is about 15.5 °C. The extreme maximum temperature recorded was about 38.8 °C on 23 April 1958, and the extreme minimum temperature was recorded at about 1.4 °C on 12 January 1955. The annual mean wind speed in the sea area of the study area is about 4.2 m/s according to the statistics of meteorological observation data over the years. The average wind speed in June is the highest up to about 4.9 m/s, and the average wind speed of about 3.8 m/s in March is the lowest. Northeast wind prevails in autumn and winter, south wind prevails in summer, and the two wind directions alternate in spring. The dominant wind direction throughout the year is northeast, with an occurrence frequency of 23%, and the secondary dominant wind direction is south, with an occurrence frequency of 20%.

Located in a typical tropical monsoon climate with high temperatures and less precipitation in the tropics, the study area characterizes relatively high seawater temperature throughout the year with an annual mean surface seawater temperature of about 26.2 °C, the lowest mean seawater temperature of about 20.2 °C in January, and the highest at about 30.1 °C in June according to the statistics of seawater temperature recorded from 1960 to 2007. The average seawater temperature increases month by month over the first half of a year, the quickest increase appears in March and April, with a temperature rising rate of about 3.1 °C/month from January to June. In the latter half of the year, the seawater temperature decreases gradually; the quickest decrease takes place in November and December, and the cooling rate reached 4.0 °C/month from July to December.

### 2.1.3. Tide Activity

Tide activities near the study site were systematically monitored and observed with airborne and in-situ equipment periodically in the past several years. For 2017 (Figure 2a), the highest tide level observed occurred at 7:07 on 7 November, with a tide height of 2.47 m, and the lowest occurred at 7:13 on 2 November, with a tide height of 0.55 m in November. For 2018 (Figure 2b,c), the observed highest tide level appeared at 03:00 on 1 March, with a tide height of 2.09 m, and the lowest appeared at 4:05 on 23 March, with a tide height of −0.17 m, over the first half of 2018. The highest tide level appeared at 16:00 on 10 September, with a tide height of 2.15 m, and the lowest appeared at 4:05 on 16 September, with a tide height of 0.07 m, which is over the latter half of 2018, according to the observed daily and hourly records of tide activities provided by Hainan maritime department.

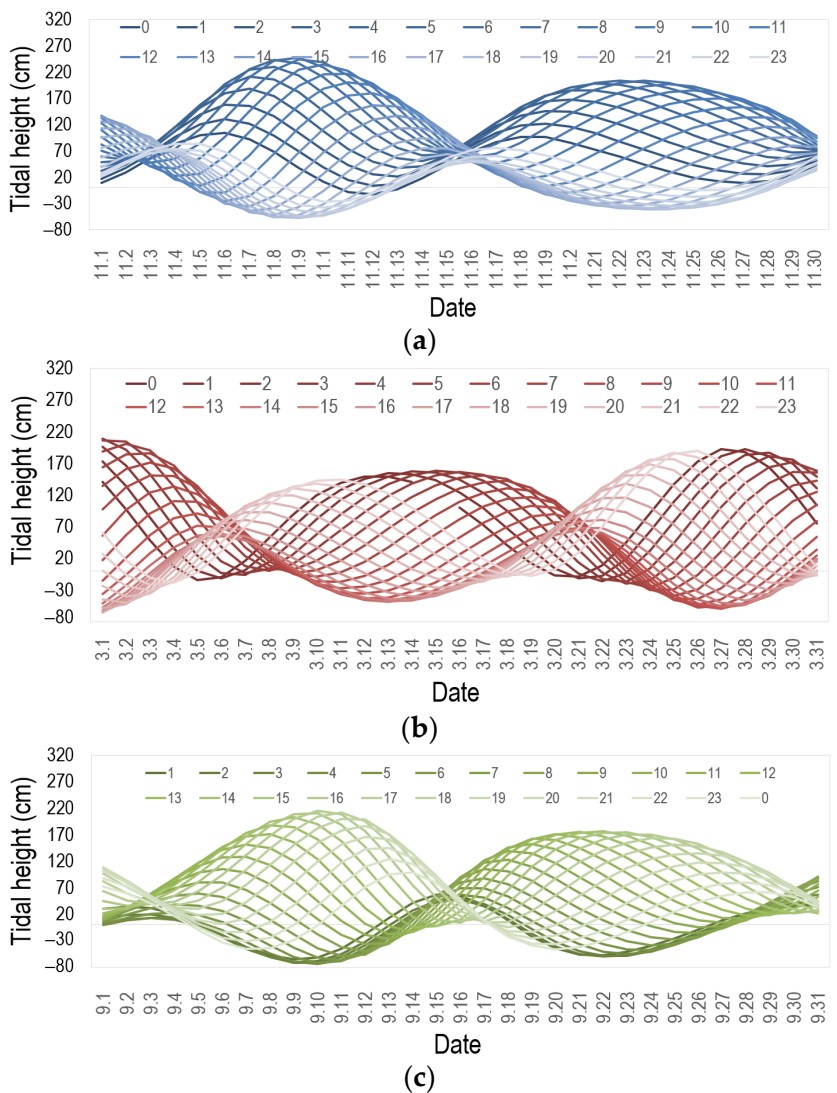

**Figure 2.** Variation diagram of tidal height of inshore waters over three periods in coastal waters. (**a**) Daily and hourly tidal height change map of inshore waters in November 2017; (**b**) Daily and hourly tidal height change map of inshore waters in March 2018; (**c**) Daily and hourly tidal height change map of inshore waters in September 2018.

### 2.2. Instruments

The airborne thermal infrared imaging spectral system TASI-600 from the Canadian ITRES company were used as the sensors for the seawater temperature drainage inversions in the present study [13]. TASI has high spectral resolution and a high signal-to-noise ratio, which is suitable for water temperature monitoring in such a complex environment as the ocean. The spectral range is 8.0–11.5 μm, the number of pixels per line is 600, the continuous spectrum of channels is 32, the spectral bandwidth is 0.125 μm, the total FOV is 40°, the signal-to-noise ratio (peak) is 4600, and the absolute radiometric accuracy is ±10%. The SBE56 water temperature measuring instrument produced by Sea-Bird Electronics consisted of a low-power, high-precision thermometer powered by an internal battery to record temperature at the fixed measuring time interval to ensure synchronous measurements with the "space-sky" observational system, was used for sea surface water temperature measurement as ground truth.

### 2.3. Airborne Hyperspectral Thermal Infrared Data

TASI-600, an airborne thermal infrared imaging spectral system introduced by ITRE [4], takes the technical advantages of both high spatial and high spectral resolution. It mainly

consists of an imaging spectrum sensor, an Instrument Control Unit (ICU) central controller, and a series of precise geometric and radiometric correction instruments. Given the temperature and emissivity of the blackbody, the radiance value at a given wavelength can be calculated with the Planck blackbody formula, and the radiation sensitivity coefficient of the sensor thus can be calibrated to derive the radiation calibration file [12,13]. After radiometric and geometric corrections, the images acquired in the study area can be mosaiced for multiple navigation belts [33]. Four airborne data acquisitions were designed, including two spring tides and two neap tides in winter and summer respectively, among which the airborne data acquisitions in the winter were arranged on 18 November 2017 and 15 March 2018, while in the summer they were arranged on 1 September 2018 and 9 September 2018, according to the latest ocean and hydrological observation data in the local sea area (Figure 3). The specific flight time is calculated according to the tide table, and a total of 8 lines are required for each time work (Table 1).

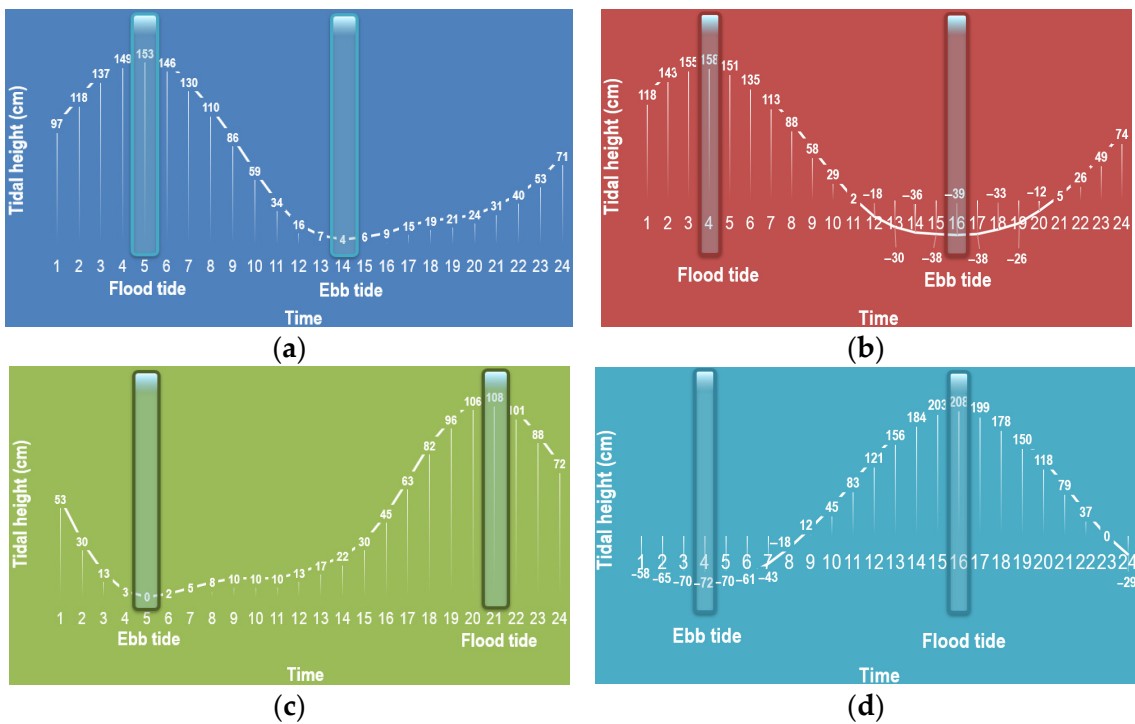

**Figure 3.** The time window of airborne thermal infrared data acquisition. (**a**) Time window for airborne data acquisition during the spring tide on 18 November 2017; (**b**) Time window for airborne data acquisition during the neap tide on 15 March 2018; (**c**) Time window for airborne data acquisition during the spring tide on 1 September 2018; (**d**) Time window for airborne data acquisition during the neap tide on 9 September 2018.

**Table 1.** Airborne remote sensing data acquisitions.

| No. | Type | Imaging Time | Spatial Resolution | Tidal Condition |
|---|---|---|---|---|
| 1 | TASI | 18 November 2017 | 1.0 m | Neap tide in winter |
| 2 | TASI | 15 March 2018 | 1.0 m | Spring tide in winter |
| 3 | TASI | 1 September 2018 | 1.0 m | Neap tide in summer |
| 4 | TASI | 9 September 2018 | 1.0 m | Spring tide in summer |

### 2.4. Sea Surface Temperature Data

The seawater temperature sensors onboard the ship were calibrated indoors in advance and set to 1 time/s to coincide with the measurement timing of a handheld Global Positioning System (GPS) for ensuring the simultaneous normal working state of the instruments

outdoors [23]. Along with moving the ship towards each synchronization point/quasi-synchronization point, the hydrometer onboard the ship continuously recorded the observed seawater temperature with the observation time. At each synchronization point/quasi-synchronization points, the ship berthed for about 1 min, waiting for the aircraft to pass the line, and the onboard hydrometer recorded the observed seawater temperature at the point; simultaneously, the GPS continuously recorded the ship's track and synchronization point coordinates with the time. The corresponding relationship between GPS coordinates and temperature data thus could be established, and the time series of seawater temperatures observed for all synchronization points with coordinates attached were ultimately obtained. Figure 4a just presented the geographical coordinates of the sea surface observation points and the corresponding measured temperature values. Figure 4b showed the SBE56 water temperature measuring instrument produced by Sea-Bird Electronics used for the present study. SBE56 is a low-power, high-precision thermometer powered by an internal battery to measure and record temperature with time. The probe of the hydrometer was fixed with a rope to ensure the probe collects the seawater temperature at about 10~30 cm below the water surface as shown in Figure 4c.

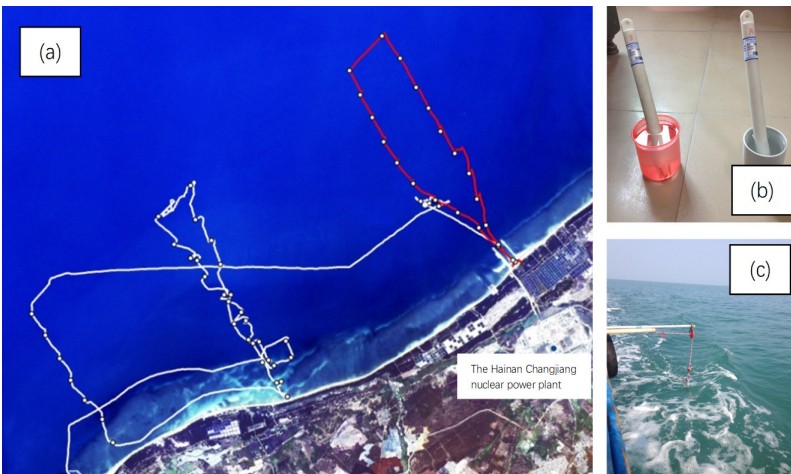

**Figure 4.** Seawater temperature measuring points along with routes of two ship field observations. (**a**) Geographical coordinates of the sea surface observation points; (**b**) SBE56 water temperature measuring instrument produced by Sea-Bird Electronics; (**c**) The probe of the hydrometer was fixed with a rope.

After the field champion, a total of 2640 measured seawater temperature data were obtained for each tide level at each measurement frequency of 1 s (Table 2). The standard deviation of the measured data was lower than 1 °C, indicating that the instrument worked normally with satisfactory performances.

**Table 2.** Statistics of the measured sea surface temperature data.

| No. | Date | Minimum Value (°C) | Maximum Value (°C) | Mean Value (°C) | Range (°C) | Standard Deviation (°C) | Tidal Condition |
|---|---|---|---|---|---|---|---|
| 1 | 18 November 2017 | 25.43 | 29.21 | 27.52 | 3.78 | 0.41 | Neap tide in winter |
| | | 26.35 | 30.01 | 28.31 | 3.66 | 0.42 | |
| 2 | 15 March 2018 | 24.33 | 27.02 | 26.03 | 2.69 | 0.38 | Spring tide in winter |
| | | 22.54 | 26.53 | 24.03 | 3.99 | 0.45 | |
| 3 | 1 September 2018 | 28.51 | 30.52 | 29.21 | 2.01 | 0.38 | Neap tide in summer |
| | | 26.39 | 30.41 | 28.24 | 4.02 | 0.63 | |
| 4 | 9 September 2018 | 26.85 | 29.63 | 28.11 | 2.78 | 0.35 | Spring tide in summer |
| | | 26.89 | 30.85 | 28.58 | 3.96 | 0.82 | |

### 2.5. Algorithm Development

The temperature inversion algorithms popularized for sea surface mainly include a radiative transfer equation algorithm [35], a single window algorithm [36], a universal single channel algorithm [37], and a Wen algorithm [38]. The atmospheric influence on sea surface thermal radiation should first be corrected according to the real-time atmospheric sounding data or standard atmospheric profile data, usually by using Moderate Spectral Resolution Atmospheric Transmittance (MODTRAN), Low Spectral Resolution Transmission (LOWTRAN) or Second Simulation of Satellite Signal in the Solar Spectrum (6S) models based on the radiative transfer algorithm. The principle is to calculate the atmospheric absorption in the heat radiation conduction and the upward and downward radiation intensity of the atmosphere itself, subtracting the atmospheric influence from the total radiation intensity observed at the height of the sensor, and finally, obtain the surface heat radiation intensity.

The Digital Number (DN) value of the acquired data by the system was first converted into the radiance value with the radiation correction software Radiometric Correction (RADCORR). The main parameters for the spectral and radiative calibrations were determined at the laboratory. The laser generator with a known wavelength was mainly used for spectral calibration. Since the wavelength of the standard light source was known, and the position of the wavelength on the sensor array was measured by energy, the central wavelength of each pixel can, thus, be estimated by the least square polynomial fitting, and the spectral calibration file was obtained. Given the relevant atmospheric parameters, the image can be corrected subsequently with an atmospheric radiative transfer equation after the radiation correction or after the geometric correction. The remote sensing image records the GPS time of the image in a time mark record file that contains the temporal records of the Position and Orientation System (POS). Through time comparison and coordinate projection transformation, the attitude data and position data of each remotely sensed image can be provided for geometric correction on the image. A High-Precision Digital Elevation Model (DEM) is necessary to improve the accuracy of geometric correction by eliminating the topographic effects. However, as for the present study area located in the sea area, the influence of terrain can be ignored. For the image mosaic, two steps, including image registration and image mosaic, are required. The former aligns the overlapping parts of the images through image transformation and "stitches" them into a new image for final image mosaicking. In this paper, we choose the Random Sample Consensus (RANSAC) algorithm which is based on point feature matching and performs excellently on TASI data. For eliminating the boundary effect that usually occurred in image mosaicking, the overlapping parts were smoothed to improve the image quality to realize seamless image stitching. Many approaches were proposed for image stitching, among which the simplest one is median filtering and weighted average fusion, and the most complex ones are the image Voronoi weighting method and Gaussian spline interpolation method. In this paper, we choose Gaussian spline interpolation.

Under the assumption that the properties of the surface and atmosphere are Lambertian for thermal radiation, the atmospheric downward radiation intensity should be constant in the hemispheric space, the thermal radiation transfer equation can be simplified as [39]:

$$L_\gamma = B(\gamma, T_S)\varepsilon\tau(\gamma) + L_\uparrow + L_\downarrow(1 - \varepsilon)\tau(\gamma) \qquad (1)$$

where, $L_\gamma$ is the thermal infrared radiance received from the remote sensor at wavelength $\gamma$; $B(\gamma, T_S)$ is the Planck blackbody radiance at the surface physical temperature $T_S(k)$; $\varepsilon$ is surface emissivity at wavelength $\gamma$; $\tau(\gamma)$ is the atmospheric transmittance from the ground to the remote sensor; $L_\uparrow$, $L_\downarrow$ are atmospheric upward radiation and atmospheric downward radiation, respectively.

The true surface temperature is corrected according to the surface-specific emissivity. The expression of surface temperature can be written as [37]:

$$T_S = \frac{K_2}{In(1 + K_1/B(T_S))} \tag{2}$$

where, $T_S$ is the ground temperature; $K_1$ and $K_2$ are the calibration constant of the sensor, which has different values according to different sensors; $B(T_S)$ is the temperature radiance of blackbody.

Band merging and geometric clipping, precise correction, radiometric calibration, land and water separation, cloud recognition, atmospheric correction and brightness temperature inversion were carried out step by step. The geometric correction parameters were set as follows: projection system WGS 84, projection band 50, and polynomial correction [8]. The cloud recognition algorithm was adopted to extract the cloud mask according to the minimum cloud DN value of each image due to the different ground objects having different reflections, and the DN value of the cloud layer was much larger than that of other ground objects [30]. The water/land separation algorithm was developed to extract the water/land boundary for the convenience of studying the temperature inversion of the water. Using the single band threshold method, according to the fact that the reflectance of the water body in the near-infrared band is lower than that of other ground objects, band 6 was selected here to determine the maximum reflectance value of the water as the threshold to distinguish the water body from other ground objects. Finally, a single window algorithm was used to retrieve the temperature near the power plant and the nearshore water temperature [14]. The split window algorithm was thought suitable for the calculation of sea surface temperature because the sea surface is relatively uniform, and the emissivity is known [26]. Through various combinations of the measured brightness temperature from two channels, the atmospheric influence and surface-specific emissivity can be corrected according to the following equation:

$$T_S = A_0 + A_1 T_{band1} + A_2 T_{band2} \tag{3}$$

where, $T_S$ is the sea surface temperature; $T_{band1}$ and $T_{band2}$ are the brightness temperatures derived from two different bands of the sensor; and $A_0$, $A_1$, and $A_2$ are the empirical coefficients, which depend on the surface emissivity and atmospheric state. The empirical coefficients obtained by different combinations are slightly different [28]. Multiple arrays were formed through the synchronous series of sea surface temperature measurement data at the same location of the image and brought into the equation group to solve $A_0$, $A_1$, and $A_2$. The advantage of the split window method is that it does not need atmospheric parameters for atmospheric corrections, and inversion accuracy is high by using multiple groups of synchronously measured temperature data at different times on different flight zones [5]. The error of the inversed sea surface temperature in this way was less than 0.3 °C compared with the in-situ observed. Table 3 and Figure 5 listed the measured sea surface temperature versus the inversed ones and the error for each measurement for the winter-spring tide.

**Table 3.** Relative error between the measured and the inversed sea surface temperature for the winter spring tide on 15 March 2018.

| Point Number | C-01 | C-02 | C-03 | C-04 | C-05 | C-06 | C-07 | C-08 |
|---|---|---|---|---|---|---|---|---|
| Measured temperature | 23.39 °C | 23.38 °C | 23.41 °C | 23.67 °C | 23.62 °C | 23.49 °C | 23.28 °C | 23.24 °C |
| Inversed temperature | 23.21 °C | 23.32 °C | 23.56 °C | 23.75 °C | 23.61 °C | 23.38 °C | 23.05 °C | 23.12 °C |
| Relative error | −0.18 °C | −0.06 °C | 0.15 °C | 0.08 °C | −0.01 °C | −0.11 °C | −0.23 °C | −0.12 °C |

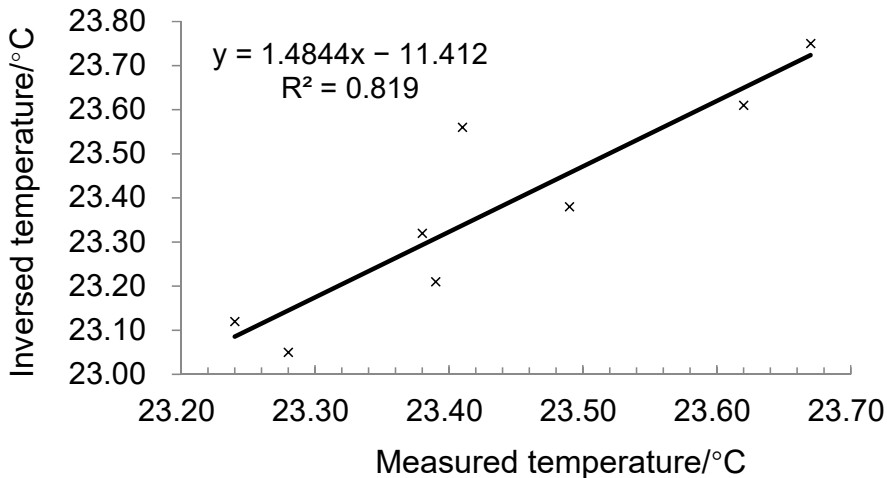

**Figure 5.** The correlation coefficient between the measured and the inversed sea surface temperature for the winter-spring tide on 15 March 2018.

### 3. Results

According to the inversed seawater temperature with airborne data, as shown in Figure 6, the following concluding remarks can be made:

(1) The seawater temperature monitored on neap tides in winter suggested that the water temperature increases gradually with the distance from the coast, which was consistent with those observed in both the flooding-type tide and ebb tide periods. Obvious temperature diffusion lines can be seen near the outlet, and the seawater temperature is nearly 1 °C higher than that of the sea area along the coastline. The water temperature in the ebb tide period is higher than that in the flooding tide period, which is 28.75 °C and 27.78 °C, respectively. In the northeast of the study area, due to the ebb of the sea during the neap tide, some shallow beaches and reefs at the base were exposed or were close to emerging over the sea, and the temperature rising effect was also obvious. The abnormally high temperature in this area was not caused by the warm drainage of the nuclear power plant only but also by the bare foreshore on the northeast and southwest coastline where the seawater temperature was high.

(2) The heating effect of the warm drainage of the nuclear power plant on the whole study area was gradually weakened with the rise of the seawater level in spring tides in winter. During rising and ebb tides, the seawater temperature near the drainage was about 24.87 °C and 23.83 °C, respectively. The flooding-type tide inundated the reefs in the northeast of the study area, making the overall temperature tended to increase gradually from the coastline to the sea. The increase in water volume in the spring tide period and the stirring effect of deep seawater exacerbated the heat dissipation effect of the warm drainage of the nuclear power plant dramatically.

(3) The monitored seawater temperature in neap tides in summer exhibited that the nuclear power plant drainage mainly affected the water temperature of the northwest sea area, so did those in the rising and the ebb tide period, and the central water temperature was approximately 29.18 °C and 27.67 °C, respectively. It was worth noting that the reef temperature in the northeast and the beach temperature in the southwest of the study area was higher than that near the drainage, but the value is lower than that in the ebb tide in winter. The reason was that the overall seawater temperature was high, owing to the high ambient temperature in summer. The water temperature tended to increase from the coastline to the drainage, and the seawater temperature near the nuclear power plant was relatively low, which was consistent with those monitored in spring and neap tide in winter.

(4) The temperature at the drainage diffused to the northeast sea area according to monitored results of spring tides in summer, which occurred in both rising and ebb tides, and the water temperature in the center of the drainage was 28.73 °C and

30.19 °C, respectively. Due to the high tide level, the temperature interference of the reef and foreshore was not obvious. An interesting phenomenon was observed, namely that an obvious temperature-rising area in the coastal area perpendicular to the discharge drainage and the coast with a relatively high range and degree of temperature rising existed, which was probably caused by a new engineering project carried out on the bank resulting in the shallower seawater and higher temperature rising after on-site investigation.

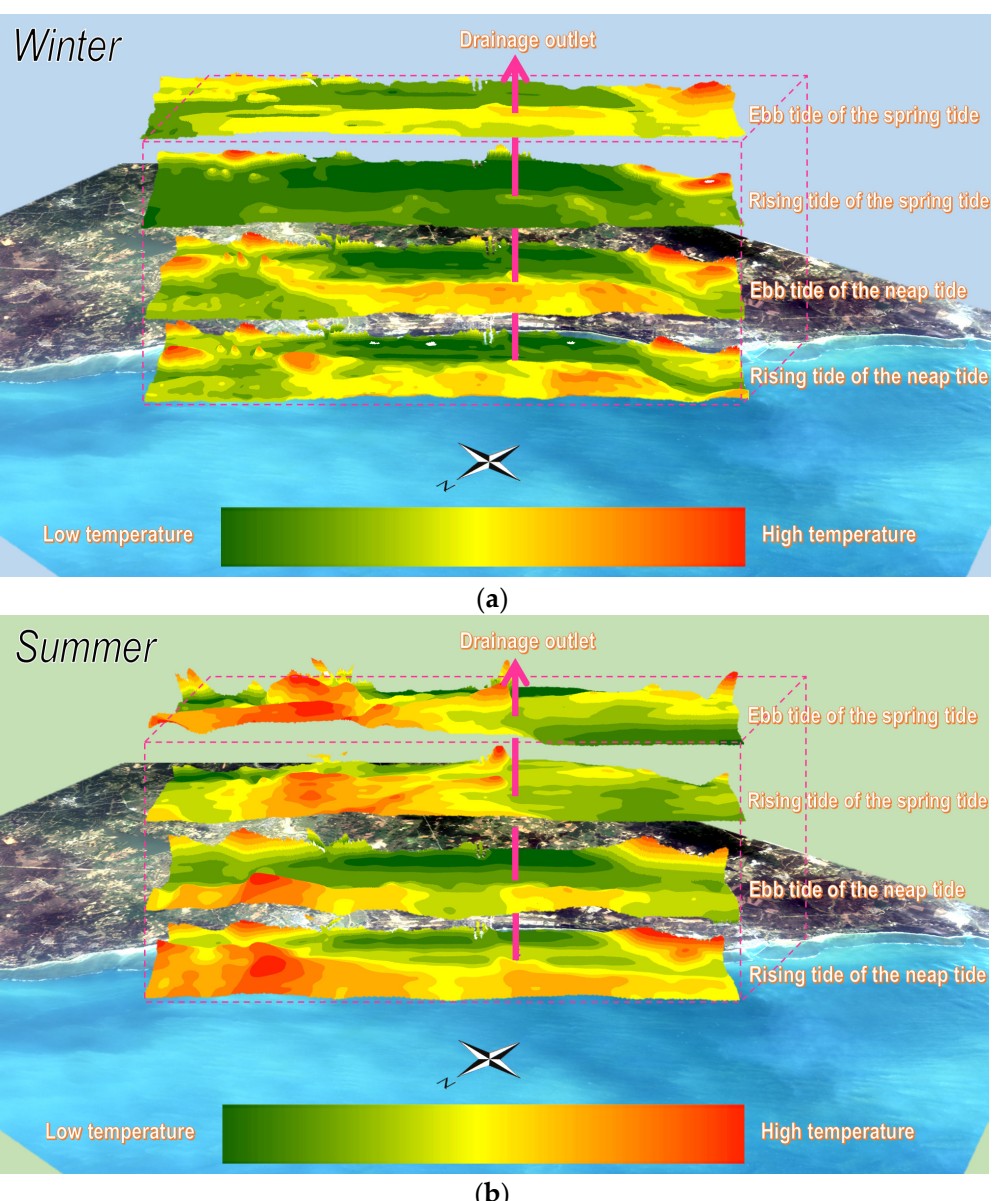

**Figure 6.** Cumulative proportion of sea area corresponding to 11 temperature levels in 3 observation periods. (**a**) Three-dimensional diagram of water temperature distribution in four tides in winter; (**b**) Three-dimensional diagram of water temperature distribution in four tides in summer.

## 4. Discussion

### 4.1. Multi-Scale Seawater Temperature Analysis

The aerial monitoring results suggested that seawater temperature varied with the distance from the coastline in certain correlations [5,23,28]. Therefore, 32 sampling points perpendicular to the coastline were designed to evaluate the accuracy of the inversed seawater temperature from multi-scale remote sensing data (Figure 7), among which No.

9 to 16 sampling points, especially No. 13 just above the drainage, needed to pay special attention for analysis.

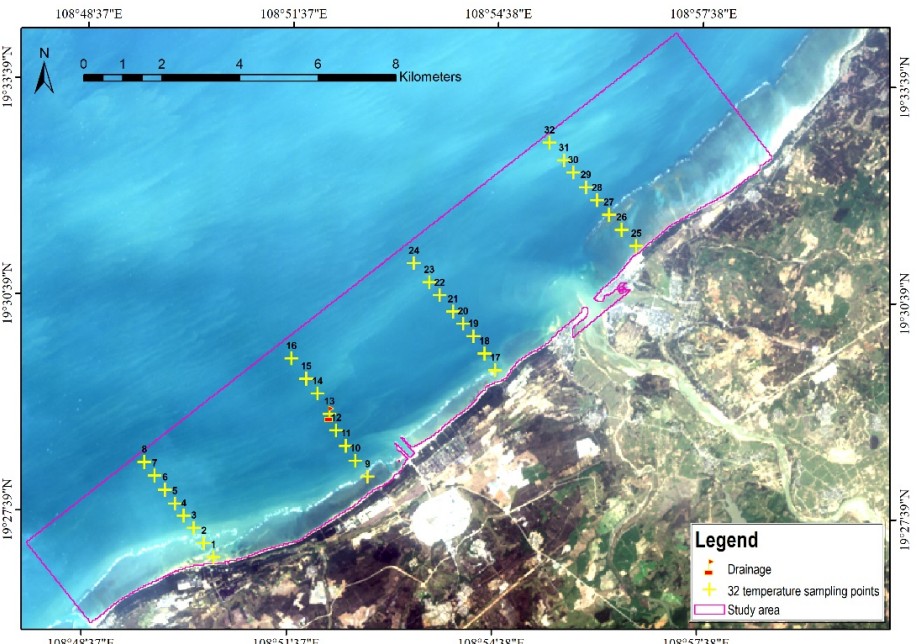

**Figure 7.** Spatial distribution of 32 temperature sampling points.

The seawater temperature fluctuated rather stably in flooding and ebb tides and was also less affected by spring and neap tides in winter and summer as presented by the red and green lines illustrated in Figure 8. In winter, the water temperature presents the opposite law at spring tide and neap tide. The water temperature at 32 sampling points is higher than the rising tide at Neap tide (Figure 8a). On the contrary, the water temperature at rising tide is higher than the ebb tide at Spring tide (Figure 8b). In summer, the water temperature distribution of rising tide and ebb tide at Neap tide is similar to that of spring tide in winter (Figure 8c). However, the tide level is generally high during Spring tide, resulting in little difference in water temperature between rising tide and ebb tide (Figure 8d). The inversed seawater temperature indicated that the water temperature at point No. 13 was not the highest among the sampling points No. 9–16. This implied that the maximum temperature rising area caused by seawater diffusion was not located directly above the drainage. Additionally, sampling points No. 9–16 exhibited a gradual cooling trend from the coastline to the deep ocean, which suggested that the dominant factor governing the seawater temperature regime may be the water depth of the sea.

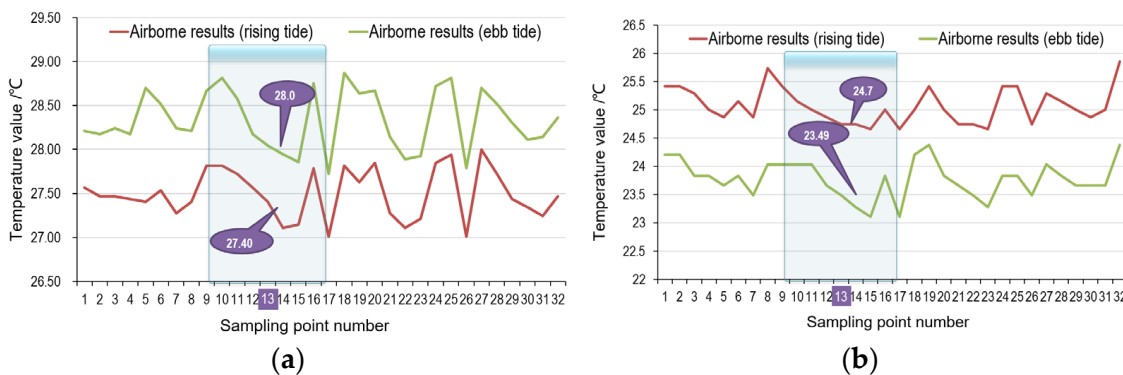

**Figure 8.** *Cont.*

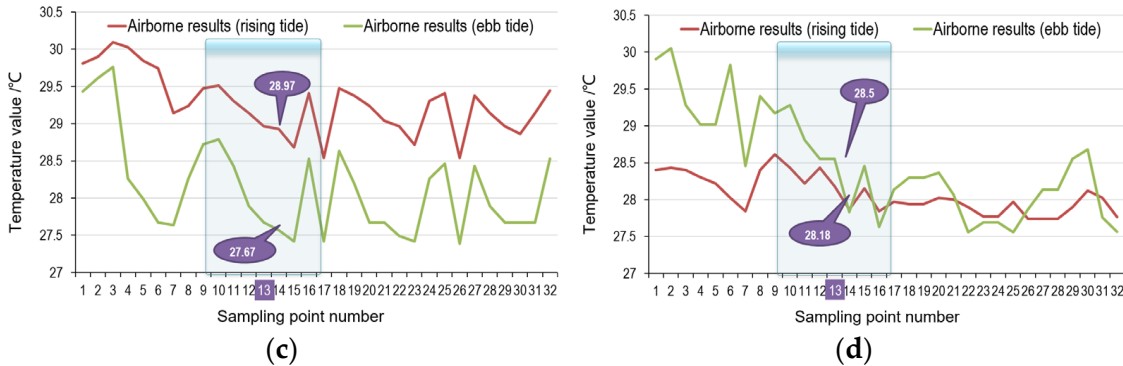

**Figure 8.** Comparisons of multi-scale seawater temperature derived from airborne observations at 32 sampling points. (**a**) Comparison of seawater temperature inversed from airborne data for Neap tide in winter. (**b**) Comparison of seawater temperature inversed from airborne data for Spring tide in winter. (**c**) Comparison of seawater temperature inversed from airborne data for Neap tide in summer. (**d**) Comparison of seawater temperature inversed from airborne data for Spring tide in summer.

### 4.2. Temperature and Tide Response

Seawater temperature is affected by mixed factors such as meteorology [26], ocean current [30], runoff [35], water depth [34], seawater convection [35], and eddy current [10] etc., and these factors are temporally and spatially differed. Comparisons of the revised seawater temperature in flooding and ebb tides during spring and neap tide between the rising and the ebb tide period indicated significant differences in absolute values, as shown in Figure 9. The effect of flooding tide and ebb tide on seawater temperature was also different, i.e., neap tides in winter (Figure 9a) and spring tides in summer (Figure 8d) exhibited similar temperature distribution patterns that the temperature in the flooding tide period was lower than that in the ebb tide period, On the contrary, spring tides in winter (Figure 9b) and neap tides in summer (Figure 9c) presented similar spatial and temporal variations, and the temperature in flood tide period was higher than that in ebb tide period in general.

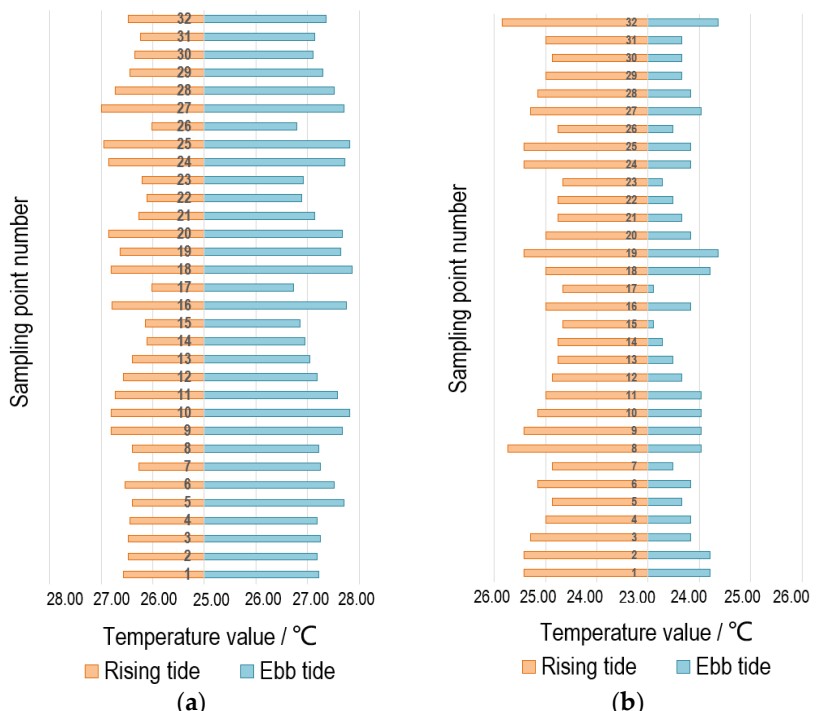

**Figure 9.** *Cont.*

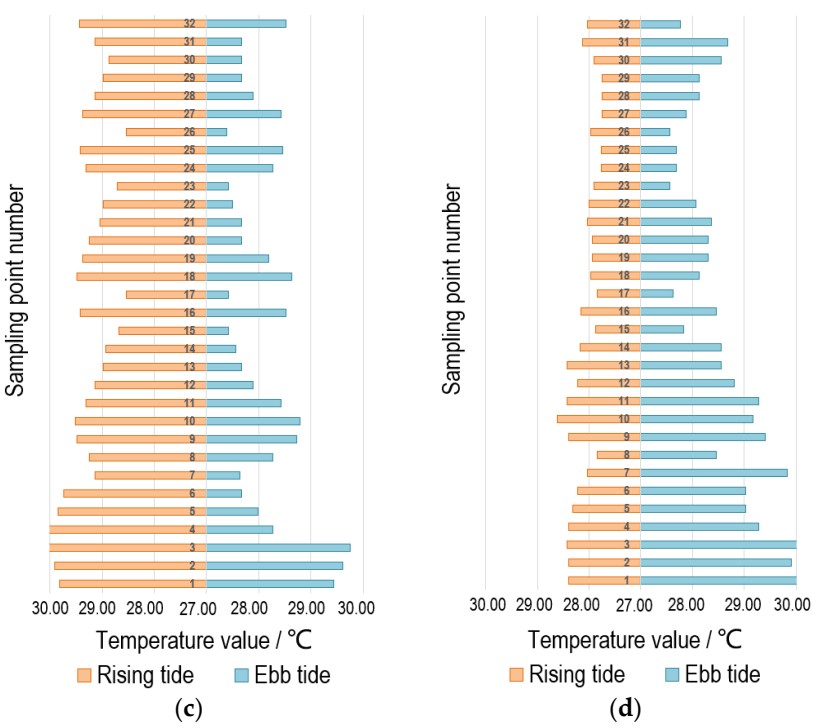

**Figure 9.** Responses between seawater temperature and tides at 32 sampling points. (**a**) Neap tide seawater temperature comparisons between rising and ebb tide in winter. (**b**) Spring tide seawater temperature comparisons between rising and ebb tide in winter. (**c**) Neap tide seawater temperature comparisons between rising and ebb tide in summer. (**d**) Spring tide seawater temperature comparisons between rising and ebb tide in summer.

Seawater temperature reversed at 32 sampling points in 8 tide periods were investigated, as shown in Figure 10, that the values of seawater temperature concentrated in certain ranges for each tide period. No obvious correlations can be investigated between the tide heights and the seawater temperatures. The relatively low seawater temperatures appeared at −0.39 m and 1.58 m tide heights, while the relatively high values were found at −0.72 m, 0 m, and 1.08 m tide heights in general. In addition, the aggregation degree of seawater temperature distribution was also not directly related to tide height, this can be identified from the shape and inclination of the red circles.

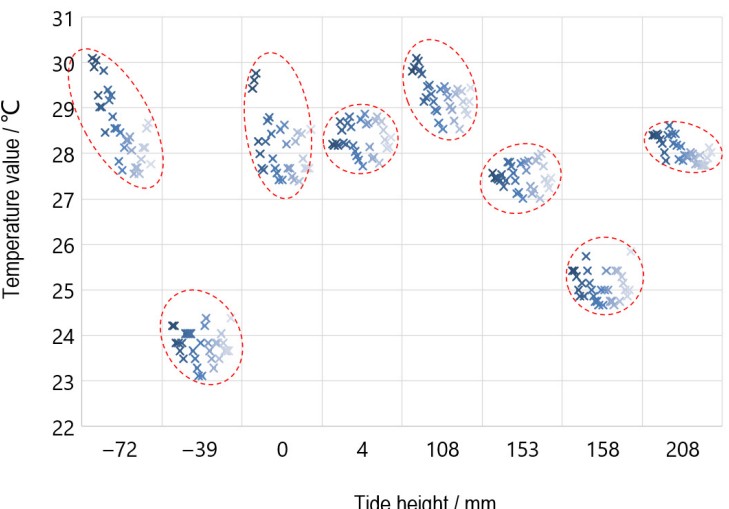

**Figure 10.** Correlation between seawater temperatures and tide heights for 32 sampling points.

### 4.3. Seasonal Variation Characteristics of Seawater Temperatures

Tides are periodic fluctuations of sea surface level caused by the seasonally varied gravitational attractions of the moon and the sun. The circle diagram of the seawater temperature in winter and in summer derived from 32 sampling points, as shown in Figure 11, exhibited unique characteristics. It can be observed clearly that the four temperature-circled lines in winter did not intersect with each other. The seawater temperature varied from high to low in the order of the ebb of spring tide, the rising of spring tide, the rising of neap tide, and the ebb of neap tide in winter, respectively. On the contrary, the seawater temperature varied slightly in the summer, and four temperature-circled lines intersected with each other, which suggested that the heat dissipation effect of warm drainage was relatively poor probably due to the high average seawater temperature in summer. Therefore, seawater temperature observed in the summer can be better used to explain the environmental impact caused by warm drainage in terms of seasonal seawater temperature variation studies. It is worth noting that the average seawater temperature in winter and summer differed by less than 9 °C, owing to the location of the study area in the tropical region.

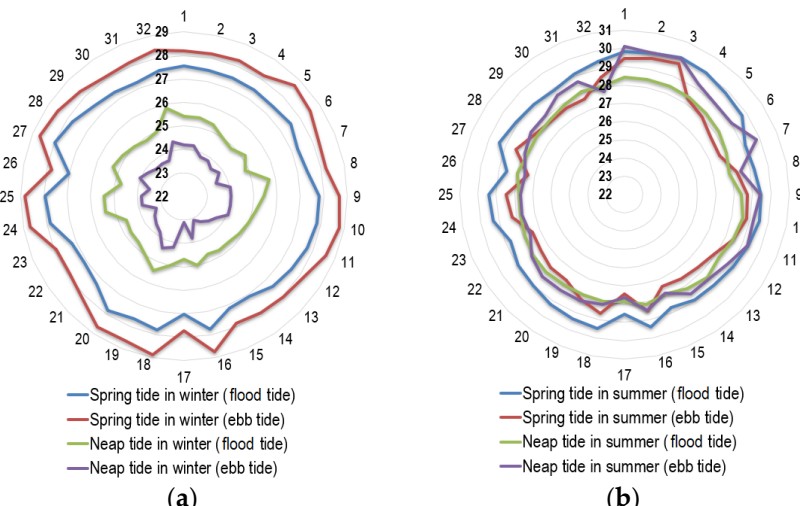

**Figure 11.** Seasonal variation characteristics of seawater temperatures derived from 32 sampling points. (**a**) The circle diagram of the seawater temperature in winter; (**b**) The circle diagram of the seawater temperature in summer.

### 4.4. Seawater Temperature Distribution Characteristics under Different Tides

The seawater temperature distribution histogram for eight monitoring periods, as shown in Figure 12, presented a variety of patterns. According to the shapes of histograms, i.e., normal type [7], island type [11], bimodal type [29], sawtooth type [26], steep wall type [27], and flat top type [31], the seawater temperature distribution characteristics can be recognized. A histogram with two peak values in shape for the rising and ebb tide of neap tide in winter exhibited a bimodal type (see Figure 12a,b), which indicated that the effect of warm drainage on seawater was rather weak. The temperature distribution histogram for the flooding tide of spring tide in winter presented a steep wall type, which was inclined to one side similar to the steep wall of a mountain (see Figure 12c), indicating that the whole sea area was not affected by the effect of warm drainage. The temperature distribution histogram for the ebb tide of spring tide in winter, rising and ebb tide of neap tide in summer presented a sawtooth pattern with uneven shape (see Figure 12d–f), which indicated that the abnormal seawater temperature was not caused by warm drainage, but by reefs and other factors near the shore. The temperature distribution histogram for rising and ebb tide of spring tide in summer exhibited an approximately symmetrical normal histogram, which was high in the middle and low on both sides (see Figure 12g,h), which suggested that seawater temperature evenly distributed, and the thermal diffusion was clear.

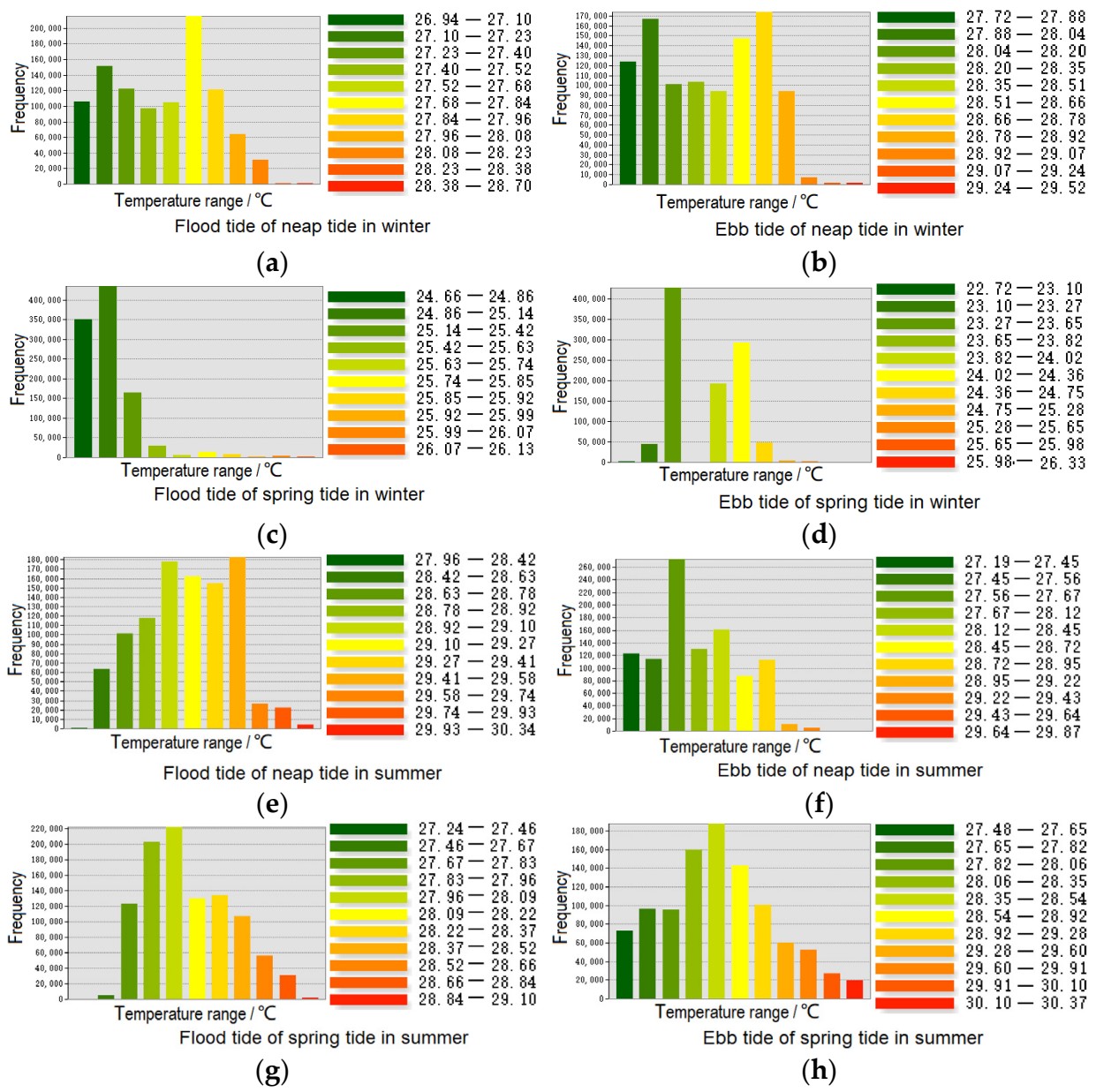

**Figure 12.** Seawater temperature distribution histograms for each tide types. (**a**) Flood tide of neap tide in winter; (**b**) Ebb tide of neap tide in winter; (**c**) Flood tide of spring tide in winter; (**d**) Ebb tide of spring tide in winter; (**e**) Flood tide of neap tide in summer; (**f**) Ebb tide of neap tide in summer; (**g**) Flood tide of spring tide in summer; (**h**) Ebb tide of spring tide in summer.

### 4.5. Exploration on the Determinants of Temperature

Theoretically, seawater temperature is affected by many factors, including season, tide level, tide state, etc. [28]. Statistical covariance analyses (see Figure 13) for each tide are very suitable for evaluating the correlation of multiple variables at any two tides. Positive covariance of seawater temperature between two tides implied that seawater temperature changes of these two tides tend to be consistent [30]. On the contrary, Positive covariance of seawater temperature between two tides implied the opposite. The water temperature of the two tides is irrelevant when the covariance is 0.

Statistical covariance analyses for each tide revealed that: (1) the highest covariance of about 0.0584 was obtained between the ebb tide of neap tide in the summer and ebb tide of spring tide in summer. It indicated that the seawater temperature between spring tide and neap tide had a positive correlation under summer ebb tide. A covariance of about 0.0443

between the flooding tide of neap tide in summer and the ebb tide of neap tide in summer ranked second, which suggested that a positive correlation between seawater temperature during flooding tide and ebb tide at neap tide in summer existed. (2) The lowest covariance of about −0.0071 was obtained between the flooding tide of spring tide in winter and the flooding tide of spring tide in summer, which indicated that under the same tide level and the same ebb and flood seawater temperature was predominately controlled by season. The second lowest covariance of about −0.0030 was found between the ebb tide of spring tide in winter and the flooding tide of spring tide in summer, which suggested that the negative correlation would be weakened when the tide level was consistent. (3) The tide level and state all showed a positive correlation in the same season, and only the water temperature in different seasons has a negative correlation. In extreme cases, such as the ebb tide of neap tide in winter vs. the flooding tide of spring tide in summer, when the season, tide state and tide level were all inconsistent, the covariance difference was approximately 0.0000, and the two become independent variables.

|  | Flood tide of neap tide in winter | Ebb tide of neap tide in winter | Flood tide of spring tide in winter | Ebb tide of spring tide in winter | Flood tide of neap tide in summer | Ebb tide of neap tide in summer | Flood tide of spring tide in summer | Ebb tide of spring tide in summer |
|---|---|---|---|---|---|---|---|---|
| Flood tide of neap tide in winter | - | 0.0239 | 0.0156 | 0.0212 | 0.0183 | 0.0256 | 0.0027 | 0.0134 |
| Ebb tide of neap tide in winter | 0.0239 | - | 0.0190 | 0.0258 | 0.0193 | 0.0248 | 0.0000 | 0.0081 |
| Flood tide of spring tide in winter | 0.0156 | 0.0190 | - | 0.0324 | 0.0151 | 0.0245 | −0.0071 | −0.0007 |
| Ebb tide of spring tide in winter | 0.0212 | 0.0258 | 0.0324 | - | 0.0198 | 0.0286 | −0.0030 | 0.0128 |
| Flood tide of neap tide in summer | 0.0183 | 0.0193 | 0.0151 | 0.0198 | - | 0.0443 | 0.0078 | 0.0376 |
| Ebb tide of neap tide in summer | 0.0256 | 0.0248 | 0.0245 | 0.0286 | 0.0443 | - | 0.0102 | 0.0584 |
| Flood tide of spring tide in summer | 0.0027 | 0.0000 | −0.0071 | −0.0030 | 0.0078 | 0.0102 | - | 0.0431 |
| Ebb tide of spring tide in summer | 0.0134 | 0.0081 | −0.0007 | 0.0128 | 0.0376 | 0.0584 | 0.0431 | - |

**Figure 13.** Covariance matrix of eight tides.

The correlation coefficients, as special covariance after standardization, for eight tides were calculated by dividing the covariance by the standard deviation, the correlation coefficients matrix was shown in Figure 14. Obviously, the covariance ranges from positive infinity to negative infinity, and the correlation coefficient can only vary between ±1. From Figure 14, one can conclude: (1) the seawater temperature data obtained on the same day have the highest correlation for the same tide in the same season. The correlation coefficients of winter neap tide, winter spring tide, summer neap tide and summer spring tide were 0.8696, 0.8459, 0.8135, and 0.7562 respectively, which implied that the tide influence was the weakest relative to the season and tide level. (2) For comparisons, groups with correlation coefficients over 0.6000 included the ebb tide of neap tide in winter vs. the ebb tide of spring tide in winter (Figure 15a), the ebb tide of neap tide in winter vs. the flood tide of neap tide in summer (Figure 15b), the ebb tide of neap tide in winter vs. the flooding tide of spring tide in winter (Figure 15c), the flooding tide of neap tide in winter vs. the flooding tide of neap tide in summer (Figure 15d) and the flooding tide of neap tide in winter vs. the ebb tide of spring tide in winter (Figure 15e), and the correlation coefficients were 0.7120, 0.6249, 0.6235, 0.6214, and 0.6153 respectively. Comparisons for these five

groups suggested that the same season (Figure 15a,c,e) and the same tide (Figure 15b,d) were the major factors determining the correlation of seawater temperature, and the effect of flooding or ebb tide was relatively weak.

| | Flood tide of neap tide in winter | Ebb tide of neap tide in winter | Flood tide of spring tide in winter | Ebb tide of spring tide in winter | Flood tide of neap tide in summer | Ebb tide of neap tide in summer | Flood tide of spring tide in summer | Ebb tide of spring tide in summer |
|---|---|---|---|---|---|---|---|---|
| Flood tide of neap tide in winter | - | 0.8696 | 0.5365 | 0.6153 | 0.6214 | 0.5284 | 0.1161 | 0.2081 |
| Ebb tide of neap tide in winter | 0.8696 | - | 0.6235 | 0.7120 | 0.6249 | 0.4889 | −0.0013 | 0.1200 |
| Flood tide of spring tide in winter | 0.5365 | 0.6235 | - | 0.8459 | 0.4625 | 0.4572 | −0.2766 | −0.0097 |
| Ebb tide of spring tide in winter | 0.6153 | 0.7120 | 0.8459 | - | 0.5096 | 0.4483 | −0.0999 | 0.1503 |
| Flood tide of neap tide in summer | 0.6214 | 0.6249 | 0.4625 | 0.5096 | - | 0.8135 | 0.2996 | 0.5164 |
| Ebb tide of neap tide in summer | 0.5284 | 0.4889 | 0.4572 | 0.4483 | 0.8135 | - | 0.2379 | 0.4896 |
| Flood tide of spring tide in summer | 0.1161 | −0.0013 | −0.2766 | −0.0999 | 0.2996 | 0.2379 | - | 0.7562 |
| Ebb tide of spring tide in summer | 0.2081 | 0.1200 | −0.0097 | 0.1503 | 0.5164 | 0.4896 | 0.7562 | - |

**Figure 14.** Correlation matrix of eight tides.

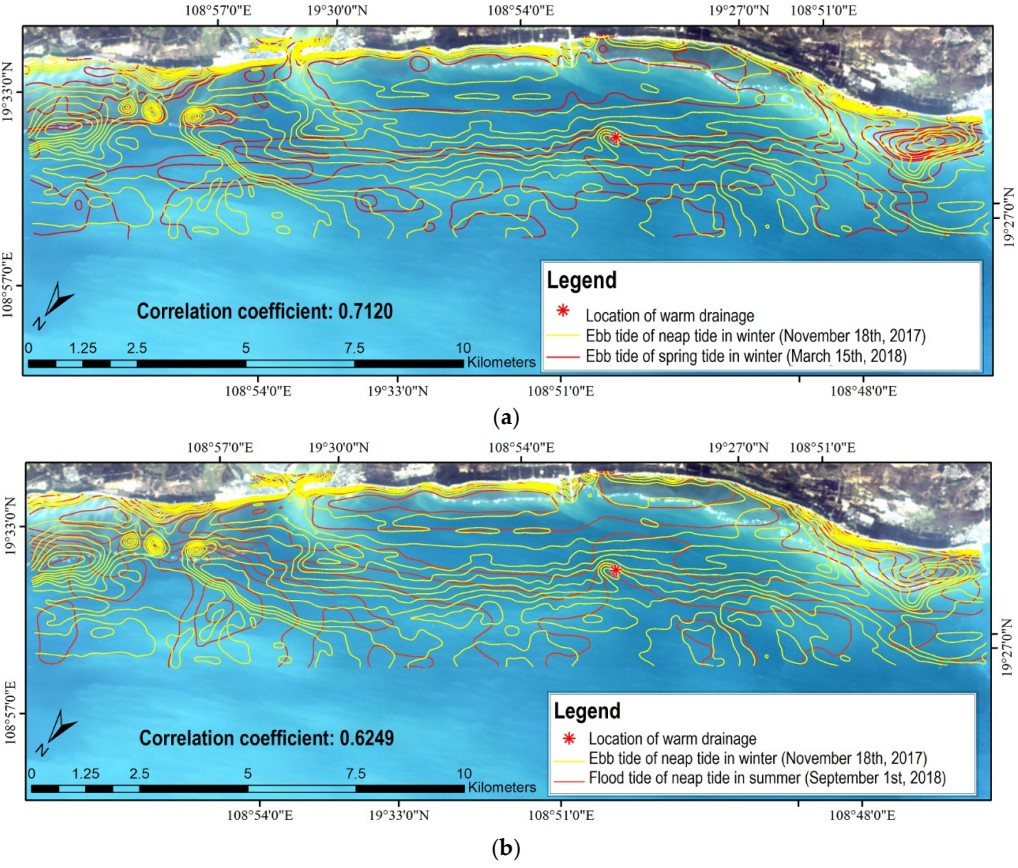

(a)

(b)

**Figure 15.** *Cont.*

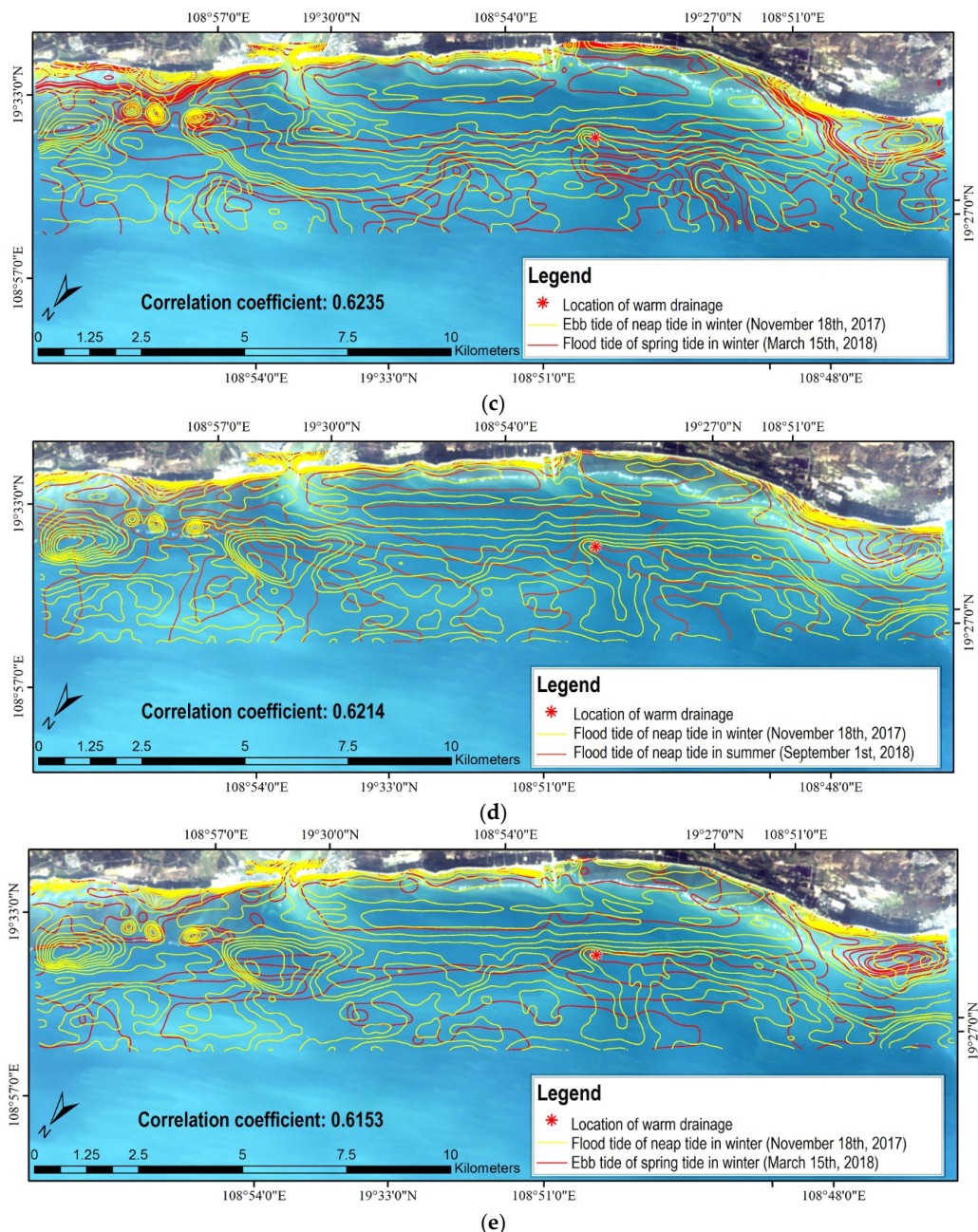

**Figure 15.** Temperature lines map with correlation coefficient greater than 0.6000 in different days. (**a**) Ebb tide of neap tide in winter vs. Ebb tide of spring tide in winter; (**b**) Ebb tide of neap tide in winter vs. Flood tide of neap tide in summer; (**c**) Ebb tide of neap tide in winter vs. Flood tide of spring tide in winter; (**d**) Flood tide of neap tide in winter vs. Flood tide of neap tide in summer; (**e**) Flood tide of neap tide in winter vs. Ebb tide of spring tide in winter.

It is concluded that season is the decisive factor of water temperature through analyses by combining the covariance and correlation coefficient of 8 tidal states. In other words, the main factor of the diffusion effect of absolute industrial temperature drainage was the overall temperature environment [3,14]. The cooler the atmosphere and the lower the ambient temperature, the more conducive to the diffusion of seawater temperature to achieve the best cooling effect; On the contrary, it will cause the warm drainage to stay in the sea area and affect the diffusion effect.

The thermal anomaly on the earth's surface mainly includes volcanic activity, thermal anomaly before seismic activity, biological combustion, industrial heat source, artificial

heat, etc. [16,25,30]. People tend to regard the temperature difference between urban and suburban areas caused by urban heat islands as a thermal anomaly, which can only reflect the overall thermal environment and cannot clearly point out the cause and specific location of thermal anomaly [35,36]. Therefore, the identification and extraction of thermal anomalies need to have a clear location, that is, to find the point with abnormally high temperature. Due to the sensor saturation problem in the thermal infrared band, many high-temperature points cannot be identified, so the mid-infrared and short-wave infrared are gradually applied to the identification of thermal anomalies [37]. It was first proposed by Dozier to measure sub pixel temperature through mid-infrared and thermal infrared channels for high temperature detection [38]. After more than 30 years of development, many changes have taken place in thermal anomaly detection from multi-spectral detection to hyperspectral detection.

## 5. Conclusions

The present study focused on analyses and discusses of the application of multi-source remote sensing technology in seawater temperature monitoring. The conclusions are as follows:

(1) The monitoring results show that the surrounding 120 km$^2$ of sea area has a warming effect due to the role of thermal drainage. The effect is most obvious in the 1 km buffer zone centered on the drainage. The seawater temperature is increased by 1.5 °C to 4.0 °C at various tide levels. Remote sensing technology provides an effective investigation means for the temperature measurement caused by drainage in industrial sea areas with its macro, fast and dynamic characteristics [12]. However, the current time resolution of aerospace remote sensing data cannot meet the measurement requirements due to the precise time required for the measurement of typical tidal conditions, such as floods, ebb periods of spring and neap tides, etc. (about 60 min). The research indicates that airborne remote sensing has the advantages of controllable measurement time and high measurement accuracy compared with aerospace remote sensing technology [35]. The implementation of airborne thermal infrared remote sensing measurement and data processing technology in this paper has important enlightening significance for the research in this field.

(2) Combined with the observation data of sea surface fixed-point water temperature according to the inversion of airborne remote sensing data of water temperature in the nearshore sea area of the plant site, the distribution map of water temperature rise area under 8 tides in winter and summer is obtained, which provided important information for the environmental impact assessment of warm drainage. The results have significant application value for mastering the response relationship between industrial warm drainage and different seasons, different tidal levels, and tidal periods, as well as the flow direction, area and distribution range of warm drainage.

(3) This study suggested that the advantages of satellite data are low cost and large-scale monitoring, but the disadvantages are that the temporal and spatial resolution is not high enough, and its application in monitoring industrial temperature and drainage is very limited. However, these shortcomings will be overcome, and the application prospect will be broad with the deployment of the constellation plan. Airborne data is flexible and can obtain multiple tidal thermal infrared data in one day. So, it is a relatively scientific data acquisition method at present.

(4) Warm drainage will cause the coastal waters to warm up, and the original phytoplankton will be wiped out massively, then bringing new species [2,3]. Consequently, a chain of ecological disasters will be triggered in the absence of natural enemies. It was concluded that the decisive factor dominating the diffusion of warm drainage was season by calculating the controlling factor of water temperature under various combinations. This conclusion has important enlightening significance for industrial drainage site selection, diffusion condition simulation and environmental impact assessment.

**Author Contributions:** Conceptualization, D.Z., Z.Z. (Zhenchang Zhu) and L.Z.; methodology, D.Z., Z.Z. (Zhenchang Zhu), L.Z., Z.Z. (Zhijie Zhang) and W.Z.; software, D.Z., X.S., Z.Z. (Zhijie Zhang), W.Z. and X.L.; validation, D.Z., Z.Z. (Zhenchang Zhu), L.Z. and X.S.; formal analysis, D.Z., Z.Z. (Zhenchang Zhu), L.Z., X.S. and Q.Z.; investigation, D.Z., Z.Z. (Zhenchang Zhu), L.Z., X.S., X.L. and Q.Z.; resources, X.S., Z.Z. (Zhijie Zhang) and W.Z.; data curation, D.Z., Z.Z. (Zhenchang Zhu), L.Z., X.L. and Q.Z.; writing—original draft preparation, D.Z., Z.Z. (Zhenchang Zhu) and L.Z.; writing—review and editing, D.Z., Z.Z. (Zhenchang Zhu), L.Z., Z.Z. (Zhijie Zhang) and W.Z.; supervision, Q.Z.; project administration, Z.Z. (Zhenchang Zhu) and L.Z.; funding acquisition, Z.Z. (Zhenchang Zhu) and L.Z. All authors have read and agreed to the published version of the manuscript.

**Funding:** This research was funded by the National Natural Science Foundation of China (No. 41830108, No. 42272346, No. 52271267), the Innovation Team of XPCC's Key Area (No. 2018CB004), Guangdong Yuehai Water Investment Co., Ltd. Multi Parameter Integrated Water Pollution Online Monitoring Technology and Demonstration Application Unveiling Project (No. JS-21-TJ-011), Forestry Innovation program in Guangdong Province (2022KJCX001) and the Major Projects of High-Resolution Earth Observation (No. 30-H30C01-9004-19/21).

**Acknowledgments:** The authors are grateful to the anonymous reviewers for their constructive comments and suggestions to improve this manuscript.

**Conflicts of Interest:** The authors declare no conflict of interest.

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
