# Peer review of "Response of Industrial Warm Drainage to Tide Revealed by Airborne and Sea Surface Observations"

_remotesensing, doi:10.3390/rs15010205_

Round 1
Reviewer 1 Report
1) General comments
This paper addresses the issue of temporal resolution using airborne remote sensing methods, retrieving sea surface temperature using a multi-band approach, and exploring the effect of industrial thermal drainage on tides from both airborne remote sensing and water surface perspectives. The subject of this paper seems interesting and the techniques used are acceptable. This study is important to investigate the effect of thermal drainage on tides in nuclear power plants. However, there are several major flaws that need to be revised and added from the method section and the results section to the discussion section and the manuscript section. Specifically, they are as follows.
First, the inversion algorithm of the manuscript lacks a description of the method and specific scatter plot results, and more importantly, the accuracy evaluation index of the results is missing, which prevents a comprehensive understanding of the overall effect and accuracy.
Secondly, the manuscript aims to explore the relationship between tidal and industrial waste water drainage, but the specific object of study is the drainage of nuclear power plant temperature, which does not seem to fit the topic well, and there is no explanation of why nuclear power plants are chosen.
In addition, in chapter 4.4, the authors provide descriptions of the shapes of various density maps, but lack specific references to supplement them, and there is a lack of literature to add to the subsequent analysis of the results for the different shapes.
As a result, I have some significant reservations about the contribution of this work to the existing body of knowledge, which led me to make the proposal for this manuscript.
To wit:
Firstly, industrial temperature and drainage has a certain effect on tides, but Hainan Province is a tropical region, the local temperature is originally high, and since this experiment is based on a tropical climate study area, the effect of industrial temperature and drainage on it may not be significant, can we add the temperature before the nuclear power plant was built and compare the before and after results.
Secondly, in 4.4, the authors provide a variety of density graphs with shape descriptions, it is suggested to add references, and also add literature to the conclusions of different shapes, and secondly, it is suggested to add the same pictures in 4.1 and 4.2 as the analysis in 4.4 to show in detail, in order to better express clearly.
Thirdly, the purpose of the manuscript is to explore the relationship between tides and industrial waste water drainage, but 4.3 only analyzes the seasonal variation of seawater temperature, whether this is out of the topic.
Finally, the writing of the manuscript required extensive editing in English, and there are many errors in grammar, spelling, and sentence structure. In the "Specific Comments" below, I provide some suggestions for improvement to improve the situation.
2) Specific comments
My suggestions are listed below. Of course, this is only a small part.
Throughout the manuscript:
1. all abbreviations in all parts of the manuscript, including the abstract and contents, as well as tables and figures, should be introduced for the first time, no matter how common the abbreviations are.
2. The algorithm section of the manuscript accounts for a very small percentage, ending after a few sentences, and the overall structure of the article has major problems.
Graphical summary
The images of the manuscript are interesting, but it is suggested to be captioned and labeled for better illustration and explanation.
Abstract
1. Most of the abstract of the manuscript is devoted to the advantages of remote sensing methods, and although these have been shown to be feasible, the authors lack an analytical description of this study, and it is suggested that the authors revise it.
2. The authors describe the data pre-processing process of TASI (band brightness and noise reduction) in the abstract, but it is missing in the text, and it is suggested that the authors add it.
3. The keywords are suggested to be adjusted to 3-5.
Introduction
1. lack of research status on UAV airborne remote sensing, the authors are invited to consult more relevant literature.
2. the authors directly use TASI sensor measurements, and suggest to explain the reason of choosing TASI.
Data and methods
The description of the algorithms is rather scarce, with no indication of the rationale for their selection, no scatter plots of specific results, and a lack of evaluation of the accuracy of the results, etc.
Results
1. Figure 5: Lack of elements such as the legend compass, while Figure 5(b) has some occlusion problems, and it is suggested to adjust the angle.
2. Because there are more analyses related to seafloor topography in the results, it is suggested to add topographic maps of the seafloor to better illustrate the results.
Discussion
1. In the analysis of 4.1 and 4.2, detailed descriptions of the picture numbers are missing, and it is suggested to be consistent with the picture descriptions in 4.4.
2. In 4.4.4, the authors provide descriptions of the shapes of various density maps, and it is suggested to include references and also add literature for the conclusions of different shapes.
3. The authors only state their own results and discussion, and lack discussion with previous studies, which is suggested to be added by the authors.
Author Response
The author accepts all the modification suggestions. See the attachment for details. Thanks again for the sincere opinions of the experts and the hard work of the editors!

Reviewer 2 Report
This paper force on the coupling relationship between warm drainage diffusion and marine environment via multi-source remote sensing thermal infrared data. Also, the reason for seawater temperature diffusion is analyzed and discussed. This paper is generally well written. However, the contents of the paper are not strongly related to the title “Response of industrial warm drainage to tide revealed by airborne and sea surface observations”. There are several points that deserve to be addressed further, and this paper needs to be reviewed carefully.
1. The four questions mentioned in line 104 to 108 “(1) To what extents… regarding to warm drainage?” have not been reasonably explained and answered, especially in the conclusions.
2. What is meaning of 8 lines of description in line 194 to 195 “The specific flight time…8 lines are required for each time work (Table 1).”?
3. Table 2 needs to add the corresponding units.
4. The Figure 5 shows the Three-dimensional diagram of water temperature distribution, it seems the temperature result from the sea bottom to surface. How was this obtained?
5. The description in line 324 to 326 “Nevertheless, the rather consistent overall fluctuation treads between the satellite and the airborne inversed…” indicates the consistency of fluctuations between the satellite and the airborne results, but it cannot be proven in Figure 7?
6. The discussion at Section 4 is mainly focuses on the temperature variations at different locations of same moment. The temperature variations over time at the same location are also worth discussing.
7. The conclusion pays more attention to the application of the multi-source remote sensing technology and does not summary the influence and effect of the industrial warm drainage to tide. This makes the paper more like a technical application analysis.
8. The quality of Figures shown in the paper cannot enable the reader to read clearly.
Author Response

(The authors gave the same response as above.)

Reviewer 3 Report
The paper presents results of a study on major factors influencing the diffusion of warm water discharged after the cooling process from the nuclear power plant into the sea water. The research was carried out with the help of selected remote sensing techniques and field observations, and the Hainan Changjiang nuclear power plant, Hainan Province, South China, was taken as the case study. In my view, this is an interesting study, and the Authors have made considerable efforts to analyse the problem. However, the manuscript requires some improvements before its final acceptance for publication in the journal. Major flaws are as follows:
1. Coastal Environment: “The annual mean temperature in the sea area nearby the power plant is about 24.9 °C, and the coldest month in a year is averaged approximately 18.8 °C in January according to the meteorological record. The temperature fluctuated slightly over whole seasons of a year with the maximum difference of about 10.5 °C. The hottest season appears in June and July of a year with annual mean maximum temperature of about 28.8 °C. The annual mean minimum temperature is about 21.8 °C, usually appears in January, the lowest temperature recorded in January is about 15.5 °C. The extreme maximum temperature recorded was about 38.8 °C on April 23, 1958, and the extreme minimum temperature was recorded about 1.4 °C on January 12, 1955. The annual mean wind speed in the sea area of the study area is about 4.2m/s according to the statistics of meteorological observation data over the years. The average wind speed in June is the highest up to about 4.9m/s, and the average wind speed of about 3.8m/s in March is the lowest.” (p. 3-4, l. 132-142). Please specify the multi-year period for which these meteorological data are given.
2. P. 6, l. 219-221: “The probe of the hydrometer was fixed with a rope to ensure the probe to collect the sea water temperature at about 10~30cm below the water surface as shown in Figure 4c.”
I am wondering if that relatively shallow depth (10-30 cm) was sufficient to analyse changes in the sea water temperature. Please explain.
3. Table 2: please add unit (°C).
4. P. 6, l. 224-225: “(…), a total of 2640 measured sea water temperature data were obtained for each tide level (…) (Table 2).”
In fact, Table 2 shows pairs of temperatures (minimum, maximum etc.). However, in my opinion, it would be useful to add an explanation below the table, to what tide level each of the given pairs of temperatures corresponds to.
5. Figures 7 and 8: please unify the axes labelling: write rather “Sampling point number” instead of “Sample number” (Fig. 7) and “Points number” (Fig. 8).
6. Conclusions: “Warm drainage will cause the coastal waters to warm up, and the original phytoplankton will be wiped out massively, then bringing new species [2,35]. Consequently, a chain of ecological disasters will be triggered in the absence of natural enemies.” (p. 16, l. 470-472). I suggest removing these sentences from the introductory part of this chapter, as the study focuses rather on the physical processes than the impact of warm drainage on the sea ecosystems. After redrafting, you may put it as one of the final conclusions at the end of this chapter, underlining the usefulness of your research for example in studies on ecology of the marine environment.
7. It is strongly recommended to enlarge the figures and improve their quality. The figures are far too small and thus unreadable, which makes the article very difficult to read and analyse its content.
8. The paper requires linguistic corrections, including grammar and orthography.
To sum up, it is recommended to accept the paper for publication after major revision.
Author Response

(The authors gave the same response as above.)

Round 2
Reviewer 1 Report
The manuscript has certainly improved after the corrections. I appreciate the authors have addressed all my comments. Therefore, I consider this manuscript, after only a few minor corrections, is suitable for publication in Remote Sensing.
Reviewer 2 Report
The authors have responsed all my questions and the quality of the paper has been improved.
Reviewer 3 Report
In my opinion, corrections made by the Authors are satisfactory. Thus, it is recommended to accept the paper for publication in present form.